# Site-specific targeting of a light activated dCas9-KillerRed fusion protein generates transient, localized regions of oxidative DNA damage

Nealia C. M. House[1], Ramya Parasuram[1], Jacob V. Layer[2¤], Brendan D. Price[1]*

**1** Department of Radiation Oncology, Dana-Farber Cancer Institute, Harvard Medical School, Boston, MA, United States of America, **2** Department of Medical Oncology, Dana-Farber Cancer Institute, Harvard Medical School, Boston, MA, United States of America

¤ Current address: eGenesis, Cambridge, MA, United States of America
* Brendan_price@dfci.harvard.edu

**Data Availability Statement:** All relevant data are within the manuscript and its Supporting Information files.

## Abstract

DNA repair requires reorganization of the local chromatin structure to facilitate access to and repair of the DNA. Studying DNA double-strand break (DSB) repair in specific chromatin domains has been aided by the use of sequence-specific endonucleases to generate targeted breaks. Here, we describe a new approach that combines KillerRed, a photosensitizer that generates reactive oxygen species (ROS) when exposed to light, and the genome-targeting properties of the CRISPR/Cas9 system. Fusing KillerRed to catalytically inactive Cas9 (dCas9) generates dCas9-KR, which can then be targeted to any desired genomic region with an appropriate guide RNA. Activation of dCas9-KR with green light generates a local increase in reactive oxygen species, resulting in "clustered" oxidative damage, including both DNA breaks and base damage. Activation of dCas9-KR rapidly (within minutes) increases both γH2AX and recruitment of the KU70/80 complex. Importantly, this damage is repaired within 10 minutes of termination of light exposure, indicating that the DNA damage generated by dCas9-KR is both rapid and transient. Further, repair is carried out exclusively through NHEJ, with no detectable contribution from HR-based mechanisms. Surprisingly, sequencing of repaired DNA damage regions did not reveal any increase in either mutations or INDELs in the targeted region, implying that NHEJ has high fidelity under the conditions of low level, limited damage. The dCas9-KR approach for creating targeted damage has significant advantages over the use of endonucleases, since the duration and intensity of DNA damage can be controlled in "real time" by controlling light exposure. In addition, unlike endonucleases that carry out multiple cut-repair cycles, dCas9-KR produces a single burst of damage, more closely resembling the type of damage experienced during acute exposure to reactive oxygen species or environmental toxins. dCas9-KR is a promising system to induce DNA damage and measure site-specific repair kinetics at clustered DNA lesions.

**Funding:** This work was supported by NIH grants CA177804 (BDP) and CA93602 (BDP). One of us (JVL) is currently employed by eGenesis, Cambridge, MA. However, JVL's contribution to the current work was solely carried out while he was an employee of the Dana-Farber Cancer Institute and was completed prior to JVL's departure from the DFCI. "eGenesis did not provide support in the form of salaries to either JVL or other workers, and did not have any additional role in the study design, data collection and analysis, decision to publish, or preparation of the manuscript. The specific roles of these authors are articulated in the 'author contributions' section.

**Competing interests:** The authors declare that no competing interests exist. Although JVL is currently employed by eGenesis, all studies reported in this submission were completed while JVL was an employee of the Dana-Farber Cancer Institute. eGenesis did not provide JVL with any financial support for salary or materials, and had no role in the study design, data collection and analysis, decision to publish, or preparation of the manuscript. This does not alter our adherence to PLOS ONE policies on sharing data and materials.

## Introduction

DNA repair is a dynamic process that requires coordination between chromatin remodelers and DNA repair enzymes to detect and access DNA lesions within the complex 3D structure of chromatin [1–3]. DNA double-strand breaks (DSB) are complex lesions whose repair requires reorganization of the local chromatin structure. This process requires exchange of histone variants H2A.Z and H3.3, chromatin reorganization by remodeling complexes, including NuA4-TIP60, INO80 and CHD3/CHD4 [4–11], as well as histone modification through e.g. acetylation [12, 13]. In addition, compact chromatin structures such as heterochromatin have reduced DSB repair efficiency and require dedicated remodeling events, such as phosphorylation of KAP1 [14], to promote access to and repair of damage in these regions [15]. These processes function together to modulate chromatin accessibility at damage sites [16, 17], so that chromatin conformation does not impede the access of the DNA repair machinery to the damaged chromatin [18–22]. This chromatin reorganization in response to DNA damage then controls the recruitment of DNA repair proteins, such as 53BP1 [23], and directs repair into either homologous recombination or non-homologous end-joining DNA repair pathways [24].

Studying repair in defined chromatin domains has relied on the use of endonucleases to create multiple DSBs, such as AsiSI [25, 26], or targeted DSBs created with I-SceI, Zinc Finger Nucleases or Cas9/gRNAs [4, 27]. AsiSI generates hundreds of unique DSBs and, when coupled with chromatin immunoprecipitation and sequencing (ChIP-seq), has been used to demonstrate that transcriptionally active regions are preferentially repaired by HR [26] and to map γH2AX spreading from DSBs [25]. Targeted DSBs have been used to identify histone modifications and patterns of histone exchange following DNA damage [4, 5, 28], while Cas9/gRNA has been used widely to monitor repair mechanism pathways and the fidelity of DNA repair [27, 29]. These approaches have provided invaluable insights into how the DNA repair machinery works in concert with the chromatin. However, these systems require inducible expression of AsiSI or Cas9/gRNA [26], transfection of the expression vector [5], or the use of protein stabilizers or nuclear exclusion (reviewed in [27]), which can lead to a time delay between expression of the enzyme and robust cutting of the target site. Further, endonucleases and Cas9/gRNA induce multiple rounds of DSB production and repair, which continues until errors accumulate at the target site and eliminates the recognition sequence. As a consequence, these approaches tend to measure bulk repair products, lack the time resolution to monitor events occurring immediately after DNA damage, and may represent responses to persistent damage, rather than acute damage-repair. Further, while endonucleases induce clean, enzymatic DSBs which can be readily resected, they lack the complexity of clustered DNA damage generated by ROS or radiation [30]. Repair of endonuclease-generated breaks may therefore differ from repair of endogenous DNA damage.

Here, we have developed a system to induce rapid, transient, site-specific DNA damage that mimics complex, oxidative lesions generated *in vivo*. For this, we used the KillerRed (KR) chromophore, a GFP-related protein isolated and modified from the hydrozoan anm2CP, which generates superoxide when activated by green light [31, 32]. This superoxide is converted to a variety of reactive oxygen species, including $H_2O_2$ and hydroxyl radicals, that can cause direct base damage and DNA strand breaks [31–33]. ROS have short half-lives and limited diffusion, and therefore react very close to their site of formation [34]. Tethering of KR to transcription activators and repressors [12, 35] or histone H2B [33] has previously been demonstrated to induce localized DNA damage, including single and double DNA strand breaks, following light activation in whole cells. Here, we created a fusion protein linking KR to the C-terminus of nuclease inactive Cas9 (dCas9), generating dCas9-KR. dCas9-KR can be targeted to any desired genomic region with an appropriate guide RNA. Targeting dCas9-KR to a

unique genomic site, followed by transient exposure to activating green light, generates local damage that is preferentially and rapidly repaired by NHEJ. Further, ChIP analysis confirms rapid and reversible changes in both γH2AX and H4 acetylation, histone modifications associated with DSB repair. Surprisingly, next-generation sequence analysis indicates that dCas9-KR-induced lesions are repaired with few detectable errors. dCas9-KR is therefore a useful model for monitoring DNA repair kinetics and endogenous DNA damage following transient production of DNA damage.

## Results

### dCas9-KR is recruited to a defined chromatin site and induces site-specific DNA damage

dCas9-KR was generated by gene synthesis (DNA2.0, CA), which fused the KillerRed chromophore to the C-terminus of Cas9$_{m4}$, a nuclease-dead Cas9 that retains gRNA-mediated DNA binding [36] (See S1 File for sequence information). dCas9-KR was then targeted to the AAVS1 locus by a specific gRNA, since this locus has been previously used by us to study DSB repair [5]. dCas9-KR was transfected in absence or presence of the AAVS1 gRNA, followed by ChIP using an antibody against the KR moiety (Fig 1A). Although dCas9-KR was efficiently expressed in cells (Fig 1B), no significant enrichment of dCas9-KR protein was detected using ChIP at the AAVS1 site in the absence of gRNA (Fig 1A). However, co-expression of dCas9-KR and the AAVS1 gRNA led to robust recruitment and localization of dCas9-KR to the AAVS1 locus, where it was retained for up to 72hrs post-transfection. dCas9-KR is therefore efficiently targeted to the AAVS1 locus.

dCas9-KR was activated by exposing cells to green light (523 nm) produced by a SugarCube LED, with light intensity measured using a standard luxometer to control for consistent light delivery (S1 Fig). Initially, we determined if light activation of dCas9-KR in the nucleus leads to widespread production of DNA damage due to ROS production. For this, cells were transfected with dCas9-KR with or without sgRNA (S1C Fig), followed by exposure to 20K lux for 10 min or 1 hr. Under these conditions, there was no significant increase in global γH2AX (Fig 1C), or reduction in cell viability (Fig 1D). Activation of dCas9-KR by light does not therefore generate either widespread DNA damage or alter cell viability by these conditions.

Next, we examined if targeting of dCas9-KR to the chromatin can induce DNA damage at the targeted site. For this, γH2AX was measured by ChIP after different intensities of green LED light exposure. In the absence of the AAVS1 gRNA, dCas9-KR did not alter γH2AX after exposure to 5K or 20K lux for 1 hr (Fig 2A). Further, addition of the sgRNA to target dCas9-KR to the chromatin (Fig 1A) did not alter γH2AX in the absence of light, indicating that targeting of dCas9-KR to the AAVS1 site does not alter local γH2AX (Fig 2A, grey bars). However, subsequent exposure to 5K or 20K lux led to a rapid, dose dependent increase in γH2AX at the dCas9-KR/sgRNA binding site (Fig 2A). To verify that the increased γH2AX after light activation of dCas9-KR depends on ROS production, cells were incubated with the H$_2$O$_2$ scavenger N-acetylcysteine (NAC). NAC pre-treatment inhibited γH2AX formation after either 10 min or 60min illumination with 20K lux (Fig 2B), consistent with DNA damage being induced by ROS. To determine if oxidative base damage was occurring at the dCas9-KR target site after activation, recruitment of the base excision repair (BER) endonuclease APE1 was measured. After 1 hr of 20K lux treatment, a modest enrichment of APE1 was detected (S2A Fig), coincident with γH2AX (Fig 2A). Surprisingly, the scaffolding protein XRCC1, which is involved in stabilizing repair factors at sites of BER and single stranded breaks (SSBs), is not detectable by ChIP after dCas9-KR activation (S2B Fig). We conclude that ROS produced by dCas9-KR after green light activation generates DNA damage and increases γH2AX.

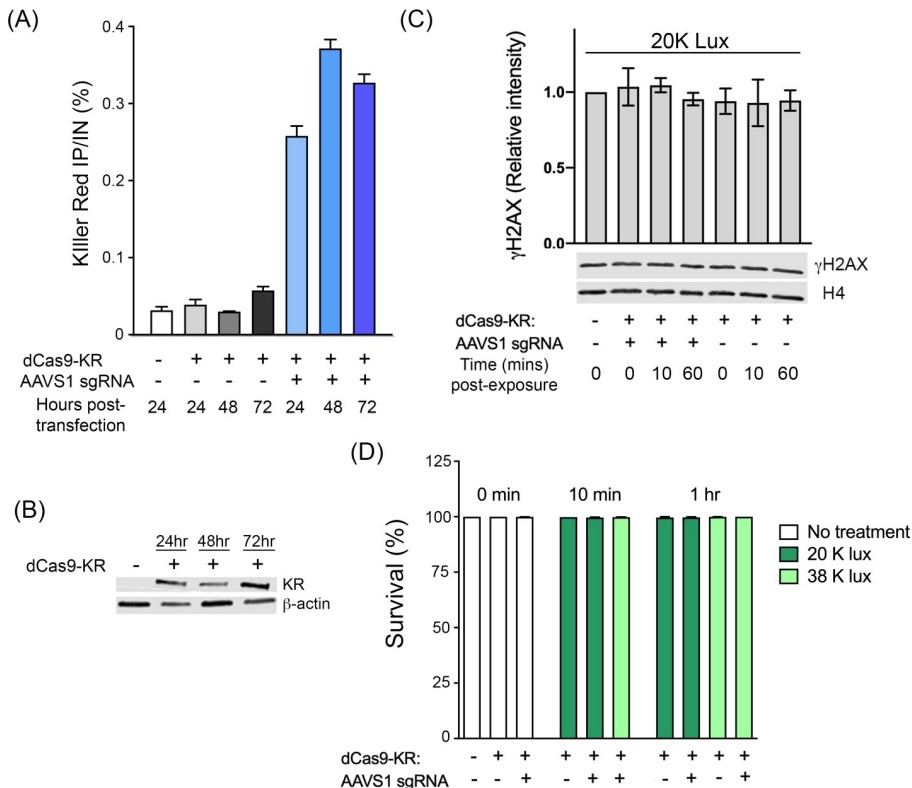

**Fig 1. dCas9-KR is recruited to the AAVS1 locus. (A)** 293T cells were transiently transfected with dCas9-KR and AAVS1 sgRNA as indicated, followed by ChIP with the KR antibody at the indicated times. Results are calculated as IP/Input signal, expressed as a percentage (n ≥ 2). **(B)** dCas9-KR expression levels in 293T cells at the indicated times following transient transfection. **(C)** 293T cells were transiently transfected with either vector (-), dCas9-KR or AAVS1 gRNA as indicated. 24 hrs later, cells were exposed to 20K Lux for the indicated times and γH2AX and H4 monitored by Western blot. γH2AX signal was quantitated using an Odyssey Imager, with 3 independent biological replicates. Statistical analysis revealed no significant increase in γH2AX under these conditions. **(D)** 293T cells expressing dCas9-KR and the AAVS1 gRNA were exposed to 20K or 38K lux for the indicated times. 24 hrs later, viable cells were measured using Trypan Blue. Percent survival relative to a paired, untreated control is plotted (n ≥ 2).

γH2AX can spread 0.5-1Mb from DSBs [25]. ROS production by dCas9-KR could generate clusters of DNA lesions that spread from the dCas9-KR binding site and could potentially be converted to DSBs. We therefore measured γH2AX spreading by ChIP after DNA damage as a marker for DSB production. Exposure to 5K or 20K lux for 1 hr led to γH2AX spreading at least 10 kb from the dCas9-KR binding site (Fig 2C). Importantly, this increase in signal was dependent on the presence of the AAVS1 sgRNA (Fig 2C, green lines). The small increase in γH2AX at -1.5kb from the dCas9-KR site (Fig 2E) is due to the low levels of H2AX occupancy at this site (S2C Fig).

Given that γH2AX was detectable after as little as 10 minutes of 20K lux exposure (Fig 2B), we next asked how rapidly repair proceeds after damage delivery. To survey repair dynamics, γH2AX was monitored during the recovery from a brief (10 minutes) exposure to 20K lux. At 1 min post-illumination, γH2AX is enriched 1.5 to 2-fold at several chromatin locations (Fig 2D). However, this increase in γH2AX signal quickly recovers and by 5 minutes post-illumination returns to baseline levels. We also examined a second epigenetic modification, the acetylation of histone H4 (H4Ac), which, like γH2AX, is rapidly increased at DSBs [5, 37]. At all loci tested, light activation of dCas9-KR immediately (within 1 min) increased H4Ac (Fig 2E). Interestingly, H4Ac was then reversed, with H4Ac dropping to levels approximately 50%

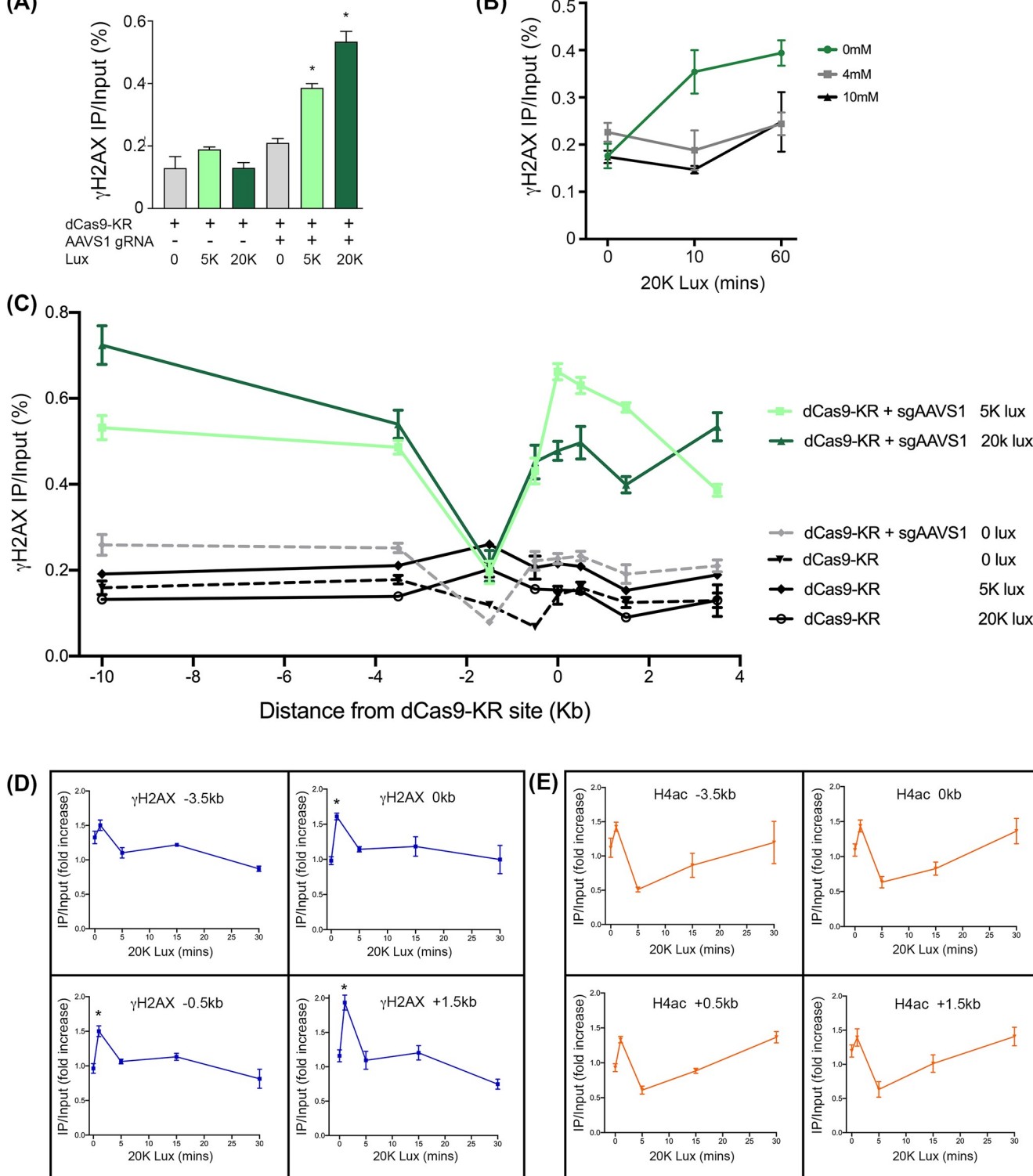

**Fig 2. ROS produced by dCas9-KR induces DNA damage. (A)** 293T cells were co-transfected with dCas9-KR in the absence or presence of the AAVS1 gRNA. 24 hrs later, cells were illuminated with green LED light at the indicated lux for 1 hr and allowed to recover at 37˚C for 15 mins, followed by ChIP for γH2AX with qPCR primers +3.5 kb from the AAVS1 gRNA site. *p values* less than 0.05 indicated as *. **(B)** 293T cells were transfected with dCas9-KR plus AAVS1 gRNA and 24 hrs later were incubated for 1 hr with N-acetylcysteine (NAC), followed by illumination with 20K lux for 10 minutes or 60 minutes. ChIP for γH2AX was carried out as in (A). **(C)** 293T cells were transfected with dCas9-KR or dCas9-KR + AAVS1 gRNA, and 24 hrs later were either not illuminated or illuminated with 5K or 20K lux for 1 hr. γH2AX ChIP was performed as in (A), using primer pairs at the indicated distance from AAVS1

sgRNA target site (0 kb). *p < 0.01* for dCas9-KR + sgAAVS1 plus 5K or 20K Lux (green lines) at all positions except -1.5kb. (**D**) Cells transiently transfected with dCas9-KR and sgAAVS1 were illuminated at 20K lux for 10 minutes and then allowed to recover for the indicated times. ChIP for γH2AX was then carried out at the indicated chromatin locations. Each experiment was paired with an unilluminated control. Enrichment of γH2AX at the indicated distances from the sgAAVS1 target site is expressed as a fold increase over NT (IP/Input illuminated divided by IP/Input not illuminated). *p values* less than 0.05 indicated as *. (**E**) As in (**D**), but with ChIP for H4ac. All ChIP experiments represent the average of at least three biological replicates with the technical SEM.

below basal values by 5 mins, coupled with re-establishment of H4Ac basal levels over the next 20 minutes (Fig 2E). Rapid increases in H4Ac followed by removal is consistent with dynamic modulation of the chromatin structure, including altered chromatin accessibility, occurring during repair [28]. We conclude that transient DNA damage generated by dCas9-KR leads to rapid chromatin modification, followed by re-establishment of the pre-existing epigenetic landscape. Further, the repair of dCas9-KR damage is rapid and essentially complete within 5 minutes of removal of the light source.

## dCas9-KR damage recruits KU70/80

The appearance of γH2AX following dCas9-KR activation (Fig 2) suggests the presence of DNA double-strand breaks (DSBs). We therefore examined if DSB repair factors were recruited to these lesions. However, neither RPA (Fig 3A) nor BRCA1 (Fig 3B) were detectable after dCas9-KR activation. To determine if persistent, unrepaired damage might be repaired by HR at later time points, dCas9-KR was activated for 10 mins, and BRCA1 monitored by ChIP during a 30 minute recovery period (Fig 3C). However, no BRCA1 was detected in this recovery/repair period. As a control, we detected robust BRCA1 accumulation when a persistent DSB was created at the dCas9-KR binding site using the p84-ZFN (Fig 3C). DNA damage created by dCas9-KR is therefore not substantially repaired by HR. We next tested recruitment of the KU70/80 complex, a key regulator of NHEJ. Exposure to either 5K or 20K lux illumination induced accumulation of KU70/80 which spread at least 10 kb from the dCas9-KR binding site (Fig 3D). Previous work indicated that KU70/80 was restricted to within 1.5 kb of the DSB [38]. The presence of KU70/80 far from the dCas9-KR site could be due to ROS diffusion from the dCas9-KR site, or reflect the 3D chromatin conformation bringing distal loci into close contact with the dCas9-KR target site, leading to widespread damage (and potentially DSBs) across the entire chromatin region. The absence of HR proteins but accumulation of the NHEJ protein KU70/80 implies that DNA damage generated by dCas9-KR is predominantly repaired via NHEJ.

## dCas9-KR damage is repaired with high fidelity

To determine if repair of dCas9-KR DNA damage led to the accumulation of mutations or insertion/deletions (INDELs), template integrity was measured by qPCR immediately after light activation, using light exposure conditions that led to accumulation of KU70/80 at the dCas9-KR site (Fig 3E). Genomic DNA remained intact after 10 minutes of 20K lux exposure, consistent with few lesions that interfere with PCR amplification (Fig 4A). Extended exposure for 1hr at 20K Lux led to variable decreases in template integrity across multiple replicates, but these did not reach statistical significance (Fig 4A). Next, we examined if this level of damage was mutagenic by sequencing DNA at the dCas9-KR binding site. Because both γH2AX (Fig 2D) and KU70/80 (Fig 3D) are highly enriched 0.5 kb on either side of the dCas9-KR binding site, we chose to amplify a 2kb product that spanned this region. DNA was isolated 24hr after exposure to either 5K or 20K lux (to allow time for repair and cell division), followed by Sanger sequencing to identify mutations. However, Sanger sequencing did not identify major

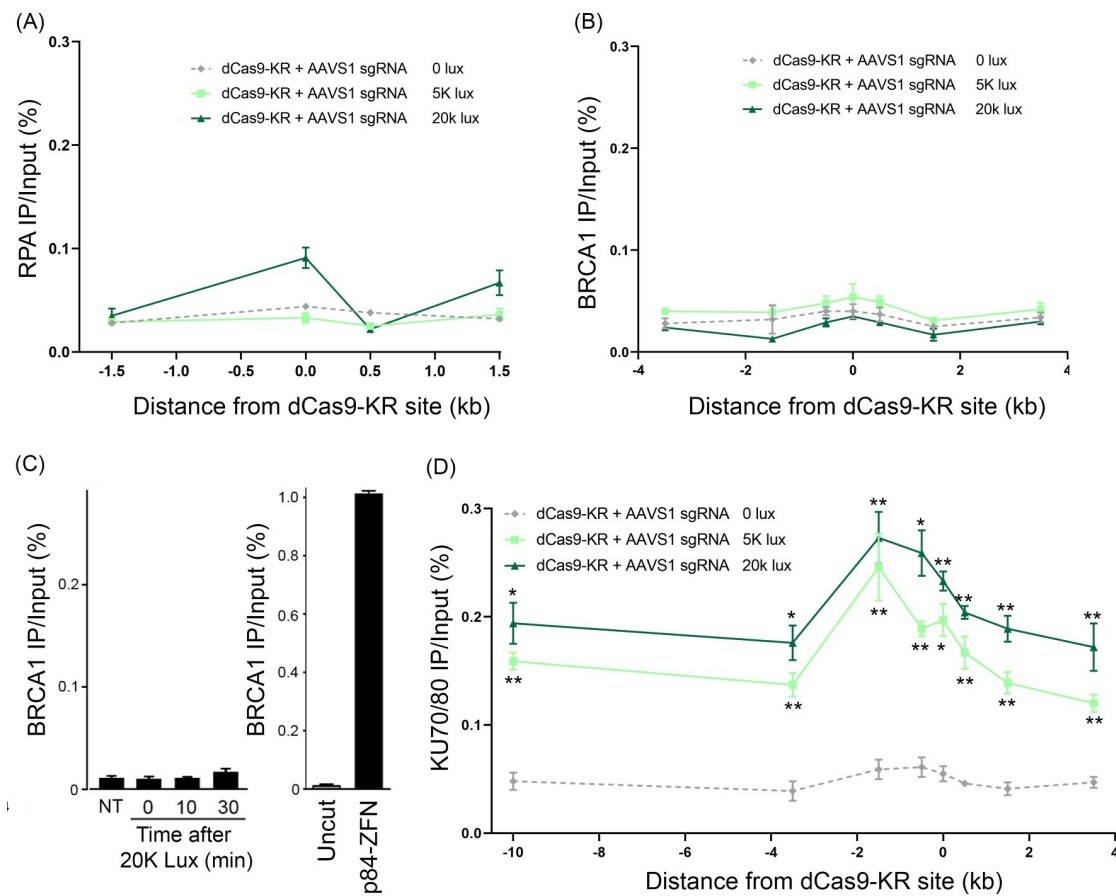

**Fig 3. The NHEJ factor Ku70/80 is recruited to dCas9-KR damage.** 293T cells were transfected with dCas9-KR and the AAVS1 gRNA, followed by exposure to 0, 5K or 20K lux for 1hr, followed by ChIP for **(A)** RPA or **(B)** BRCA1, at the indicated chromatin locations. **(C)** *Left image*: 293T cells transfected with dCas9-KR and the AAVS1 gRNA were either untreated (NT) or exposed to 20K lux for 10 min, and then either processed immediately for ChIP (t = 0 min) or allowed to recover for 10 min or 30 mins, followed by ChIP for BRCA1. *Right image*: ChIP for BRCA1 at the AAVS1 site 18hr after transfection of vector (Uncut) or p84-ZFN to generate a DSB. **(D)** 293T cells were transfected with dCas9-KR and the AAVS1 gRNA, exposed to 0, 5K or 20K lux for 1hr, followed by ChIP using an antibody specific to the KU70/80 heterodimer. All ChIP experiments represent the average of at least two biological replicates with the technical SEM, with * = p< 0.05 and ** = p< 0.01.

sequence differences between control and light exposed DNA sequences (Fig 4B). Further, the chromatographs did not indicate a substantial increase in background peaks after damage (Fig 4B), indicating that any potential INDELs or mutations occur at very low frequency. To capture what might be a low percentage of cells exhibiting mutations after dCas9-KR activation, next-generation sequencing was used to further analyze potential mutations/INDELs. dCas9-KR was activated at 20K lux for 10 min or 1 hr, allowed to recover for 48 hr, and a 241 bp genomic fragment spanning -80 bp to +161 bp at the dCas9-KR target site was amplified. Surprisingly, the mutational frequency was not increased compared to either non-transfected or dCas9-KR only (no gRNA) controls (Fig 4C; S3A–S3C Fig), even in the absence of DNA Ligase IV, a ligase essential to NHEJ (S3A–S3C Fig). There is no significant decrease in cell viability under these conditions (Fig 1), indicating that damaged cells are not lost during analysis. For comparison, cutting with either nuclease active Cas9 and the AAVS1 gRNA or p84-ZFN, which target the same sequence as the dCas9-KR/gRNA construct, significantly increased mutations at this site (Fig 4C). A Southern blot was used to attempt to capture chromosome fragility after dCas9-KR activation, but no chromosome breakage was detectable after

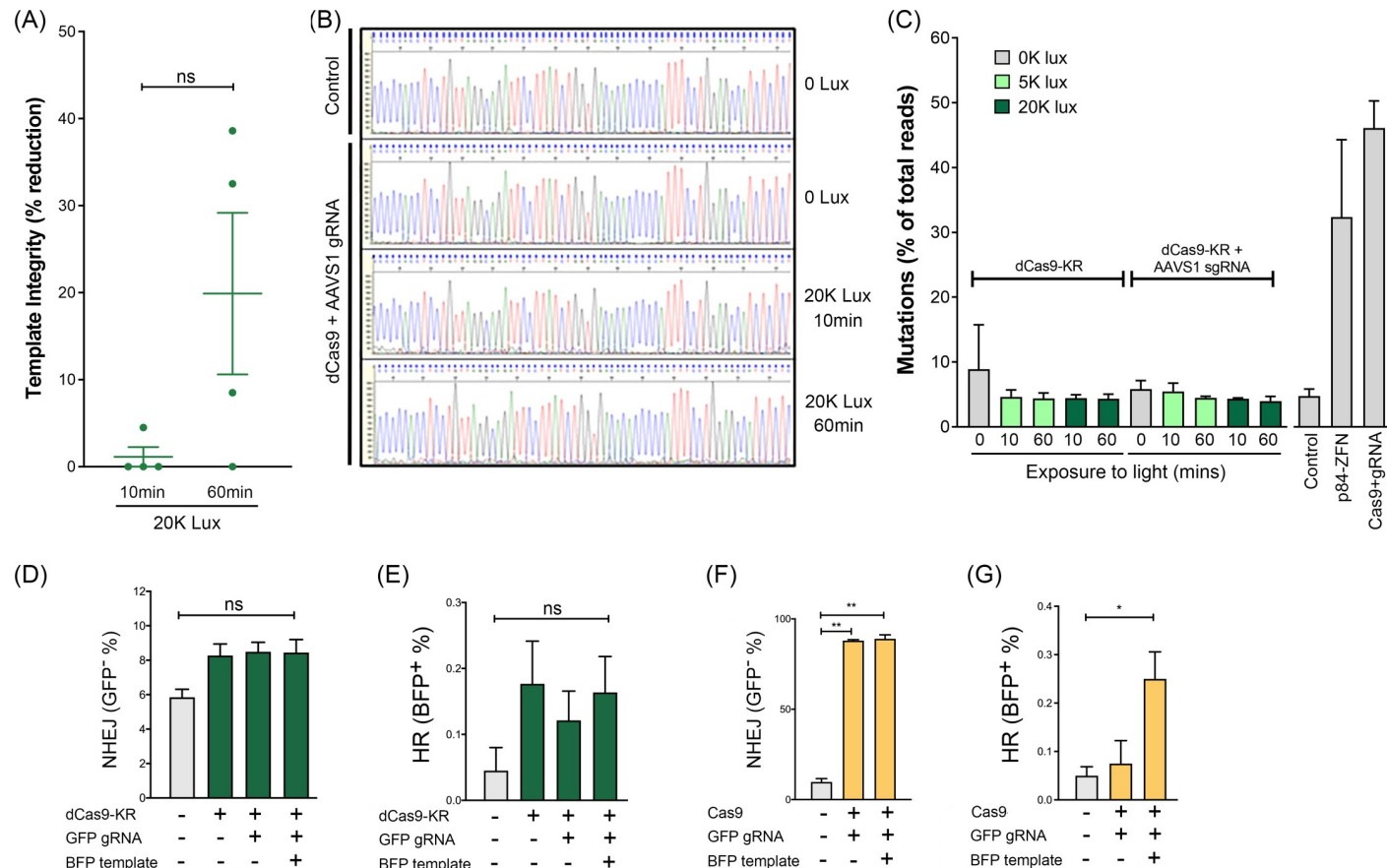

**Fig 4. dCas9-KR induced DNA damage does not increase INDELs or mutations. (A)** dCas9-KR cells were illuminated at 20K lux for 10 min or 1 hr and DNA isolated 15 minutes post-illumination. qPCR was then used to estimate the percent of intact template after dCas9-KR activation. Template integrity after illumination is expressed as a percent decrease in the quantified qPCR signal from paired, unilluminated cells. **(B)** 293T cells transiently transfected with dCas9-KR + sgAAVS1 were illuminated for 10min or 60min at 20K lux; an unilluminated control and a non-transfected control were used as a reference. DNA was isolated at 24 hr post-illumination. Representative Sanger sequencing traces are presented. **(C)** Cells transiently transfected with dCas9-KR with or without sgAAVS1 were illuminated with 0, 5K, or 20K lux for 10 minutes or 1 hr. DNA was isolated 48 hr post-illumination and a 241 bp amplicon surrounding the AAVS1 site was used for NGS. The percent of total reads that were mutated from the untreated control is plotted. Nuclease proficient Cas9 + AAVS1 gRNA and p84-ZFN included as positive controls. **(D)** Conversion from GFP⁺ to GFP⁻ to measure NHEJ after dCas9-KR ROS-induced DSB repair in untreated (0 lux) or illuminated (30K lux, 1 hr) cells. The percent of GFP⁻ cells that underwent NHEJ is plotted. **(E)** As in (D) but with addition of BFP template to monitor conversion of GFP to BFP⁺ by HR frequency after dCas9-KR ROS-induced DSB repair. **(F)** Conversion from GFP⁺ to GFP⁻ to measure NHEJ after transfection of nuclease proficient Cas9 plus GFP sgRNA to generate a DSB. The percent of GFP⁻ cells that underwent NHEJ is plotted. **(G)** As in (F) but with addition of BFP template to monitor GFP-to-BFP conversion by HR. Nuclease proficient Cas9 plus GFP sgRNA was used to generate a DSB. For (D-G), the mean and error for at least three biological replicates are plotted. * p < 0.05 and ** p< 0.005; ns = no significant change.

illumination (S3D Fig). Thus, although both γH2AX and KU70/80 are detectable immediately after dCas9-KR induced DNA damage (Figs 2 and 3), this does not lead to a detectable increase in mutations or INDELs on the repaired DNA.

## dCas9-KR DNA damage does not induce gene conversion events

To assess mutational frequency using a genetic assay, we took advantage of the GFP to BFP conversion assay in which repair of Cas9/sgRNA induced DSBs by HR and NHEJ can be measured in a single assay [39]. In this assay, DSBs generated by Cas9 in GFP which are repaired by error-prone NHEJ results in loss of GFP signal. However, when given a template for homology-mediated repair containing a missense mutation to BFP, repair by HR leads to GFP to BFP gene conversion. Because dCas9-KR produces complex damage, including base damage

and strand breaks, any mutagenic repair (e.g. via NHEJ or BER) which inactivates GFP will be detected. In addition, because this assay measures GFP+ and BFP+ cells by flow cytometry, events that occur in less than 0.1% of cells are detectable. We used the GFP-targeting gRNA with either dCas9-KR and light (Fig 4D and 4E) or nuclease proficient Cas9 (Fig 4F and 4G). Light activation of dCas9-KR did not increase error-prone repair by either NHEJ (Fig 4D) or HR (Fig 4E), even when BER was transiently inhibited by olaparib treatment to prevent repair and promote DSB formation [40, 41] (S4 Fig). In contrast, use of nuclease active Cas9 plus gRNA increased repair by both NHEJ (Fig 4F) and HR (Fig 4G). However, the GFP/BFP assay only reads out error prone repair that leads to INDELs or mutations that inactivate the GFP gene [39]. The failure to detect activity with dCas9-KR indicates that DNA damage is repaired with high efficiency/fidelity, consistent with the sequencing analysis (Fig 4C). Further, in this assay Cas9 plus the sgRNA can generate multiple cut-repair cycles over the time-course of the experiment (48 hrs), increasing the probability of repair errors, whereas transient light exposure provides a limited, temporally restricted period of damage (minutes) which is repaired with higher fidelity (Fig 4C).

## Discussion

We have developed a system which generates clustered DNA damage at a single locus to study the site-specific DNA damage response. For this, we created a fusion protein between the nuclease-inactive Cas9 and the chromophore KillerRed, dCas9-KR, allowing us to target the protein to a specified locus using specific gRNAs. Exposure to green light activates the KR moiety of dCas9-KR, generating ROS, which, in turn, cause DNA damage that is restricted to the surrounding DNA. Further, pairing this with ChIP reveals the spatio-temporal repair dynamics on the chromatin in response to DNA damage. A major advantage of this system is that generation of DNA damage can be precisely regulated by exposure to green light, allowing for transient and coordinated generation of DNA damage in all exposed cells. This provides the flexibility to monitor the early (10–20 min) events which occur during DNA repair. A second advantage of the dCas9-KR system is that, because ROS have a short (nanoseconds) half-life in cells [42, 43], termination of light exposure quickly removes the source of DNA damage, providing the ability to monitor repair in the absence of ongoing damage. Other approaches to generating DSBs with targeted nucleases, including Cas9 [27], have limitations due to the time taken for induction or expression of the enzyme, which can be 1–5 hr [44]. Further, because these approaches usually lead to constitutive expression of the nuclease, cells engage in multiple cut-repair cycles, until errors accumulate that eliminate the target site [29, 45]. Approaches that use rapid induction or degrons [46, 47] to control nuclease levels, or chemical caging of the guide RNA for CRISPR/Cas9 (vfCRISPR) [48] have provided more controlled alternatives, but still generate multiple cut-repair cycles which resemble persistent, unrepaired DSBs. Therefore, the major advantage to using the KillerRed chromophore compared to endonucleases is that ROS production is strictly limited to the duration of activation by light, allowing temporal control of DNA damage by controlling light exposure.

KillerRed has previously been used to monitor repair of oxidative damage. For example, KR was fused to the Tet repressor, allowing KR targeting to a repetitive TetR binding module which was inserted into the cells [35]. This demonstrated that BER factor recruitment is influenced by chromatin structure at the initiating lesion [35]. A similar experimental approach was taken by tethering KR to LacR to visualize CHD6 recruitment after oxidative damage [49] and tethering KR to TRF1 to localize oxidative damage to the telomeres [50]. These approaches require either expression of engineered, exogenous repeats in the cell [35, 49] or loading of the KR-TRF1 fusion onto telomeric repeats [50]. This leads to loading of hundreds of copies of the

KR construct, generating large regions of oxidative damage. However, these approaches have two key limitations which our approach overcomes. First, the need to introduce repeat cassettes limits the ability to target DNA damage to specific regions, or to survey repair in multiple cell lineages. Second, the repetitive structure of the target sites severely limits the use of e.g. sequencing to identify potential mutations or indels arising during repair. Therefore, our system allows interrogation of the repair fidelity of site-specific, clustered oxidative DNA lesions that are repaired with few errors. Viewed as a model of site-specific, clustered oxidative DNA damage, the dCas9-KR system may be used to elucidate the contribution of specific repair factors to repair outcome and probe the effect of chromatin context or DNA sequence features to repair outcome.

Previous studies have established that KR generates reactive oxidative species that directly damage the DNA [35, 49, 50]. KR-induced damage shows overlap with endogenous DNA damage arising from reactive oxygen species (ROS) that naturally occur during cellular metabolism or from exposure to oxidative agents, including oxidized bases, abasic sites, oxypyrimidines, oxypurines, and single strand breaks [12, 33, 35, 49–56]. Activation of dCas9-KR produced ROS that locally damaged the DNA, resulting in γH2AX and KU70/80 spreading at the dCas9 target sequence (Figs 2D and 3D), suggesting that DSBs result after clustered ROS delivery. Recruitment of the BER glycosylase APE1 is also detectable, though modestly. Typically, oxidized base lesions are repaired via base excision repair (BER) in which a single damaged base is excised and replaced. However, if multiple lesions occur within a ~20 bp region, this oxidative clustered DNA lesion is repaired via long-patch BER in which 2–15 bases are removed and replaced [57, 58]. If BER is inefficient or long-patch BER occurs on opposing DNA strands, DSBs can arise [59–61], though this is limited by the local chromatin structure [62]. Clustered DNA lesions induced by oxidative stress can be repaired via NHEJ, suggesting the appearance of DSBs [63]. Indeed, clustered oxidative lesions are challenging to repair, engage multiple repair pathways, and are mutagenic [30, 51, 63–68]. The coincident recruitment of APE1 and KU70/80 supports that both BER and NHEJ are occurring at sites of ROS-induced DNA damage, consistent with repair of clustered oxidative lesions requiring coordination between DSB repair and BER [63, 69, 70].

Interestingly, though DSBs do appear to be induced, mutational analysis by next generation sequencing revealed few errors at the dCas9-KR target site (Fig 4C). dCas9-KR ROS induced clustered oxidative damage is therefore repaired both efficiently and with high-fidelity. Since DSBs are arising in such low numbers, even if the initial repair via NHEJ is mutagenic, the frequency may be too low to capture by sequencing analysis. Further, if NHEJ of the DSBs is followed by BER, a high-fidelity repair event mediated by Polβ gap fill-in [71, 72], this could additionally explain the low mutational frequency captured by our analysis. Given estimates that cells experience 50,000–200,000 oxidized bases per day [73, 74], but rarely accumulate mutations, repair of low level, transient oxidative damage via BER and/or NHEJ is likely to proceed with high fidelity under limiting, "physiological" DNA damage loads delivered by dCas9-KR.

Since dCas9-KR induced damage appears to activate both NHEJ and BER, sites of dCas9-KR-induced DNA damage may represent a good model for types of DNA damage that invoke multiple repair pathways, with the ability to study repair dynamics in a site-specific manner. One caveat to the dCas9-KR system is that it does not induce as robust a DNA repair response as the systems that use e.g. targeted nucleases and therefore subtle DNA repair effects may be lost. However, we consider this to be an advantage in that it allows us to measure the DNA repair response to modest amounts of DNA damage. While IR-induced lesions take on the order of hours to resolve [59, 75–77], dCas9-KR induces rapid repair and low levels of damage are repaired within 5 minutes after light delivery is removed (Figs 2D and 3F). This suggests that the dCas9-KR induced ROS are producing lesions that are more easily repaired

than IR-induced lesions, and may be more comparable to endogenous sources of DNA damage, such as metabolites and stalled replication forks caused by DNA secondary structures. Utilizing the dCas9-KR induced ROS can add to our understanding of how these lesions are repaired with high fidelity, and therefore our understanding of how when repair goes awry it can lead to mutations.

## Conclusions

We conclude that the dCas9-KR activation described here results in targeted, clustered oxidative lesions that induce dynamic chromatin modifications and are repaired with few mutations. Pairing this technique with repair enzyme inhibitors is a promising means to elucidate site-specific and temporal responses to DNA damage. Further, dCas9-KR could be used to study repair within the context of various genomic loci–for example, to determine if the repair response is altered by transcriptional status or functional activity. dCas9-KR can also be used to explore how fragile DNA secondary structures, including repeats and G4 quadruplexes, affect repair pathway choice. A key advantage of dCas9-KR is the ability to evaluate the timing of repair factor recruitment and identify repair proteins which act on clustered DNA lesions to prevent mutagenesis. The low level of baseline mutations induced by dCas9-KR activation also make this an ideal system for further genetic perturbation, as even modest effects on repair fidelity will be detectable.

## Materials & methods

### Transfections

293T cells (purchased from the ATCC, VA) were transiently transfected with 2.5 μg pJ609-dCas-KR-puro (synthesized by DNA2.0; see S1 File for DNA and protein sequence) and/or sgAAVS1 ([78]; Addgene 41818) with Lipofectamine 2000 according to the manufacturer's instructions (Life Technologies). dCas9-KR is available from Addgene (158478).

### Light exposure

Cells in Dulbecco's Modified Eagles Media (DMEM) containing Fetal Bovine Serum (10%) and penicillin-streptomycin (1%), but without either phenol red or L-glutamine were exposed to LED light from a SugarCUBE LED Illuminator with a green bulb (Nathaniel Group/Ushio, CA) attached to a liquid light guide affixed 27cm above the plates (S1 Fig). Light intensity was measured with a luxometer placed under the plate. After exposure, cells were either placed back at 37˚C for a recovery period or immediately processed.

### Chromatin immunoprecipitation

293T cells transiently transfected with dCas9-KR in the presence or absence of sgAAVS1 were exposed to green LED light for 10 minutes or 1 hr and recovered at 37˚C for the indicated times. ChIP was performed in a minimum of three biological replicates using the following antibodies: anti-γH2AX (Abcam ab81299), anti-KU70/80 (Neomarkers MS-286-P1), anti-BRCA1 (Abcam ab16781), anti-APE1 (Abcam ab194), anti-XRCC1 (Abcam ab1838), anti-RPA (Millipore NA19L/Abcam ab2175), anti-KillerRed (Evrogen AB961). All ChIP was performed using the SimpleChIP Enzymatic Chromatin IP kit with magnetic beads according to the manufacturer's instructions (Cell Signaling Technologies). DNA levels were quantified by qPCR amplification using Power SYBR Green Mastermix (Applied Biosystems) and primers along the PPP1R12C gene (previously described in [38]; S1 File). The average IP/IN, expressed as a percent, of the biological replicates is graphed using PCR technical replicate error.

## N-Acetylcysteine treatment

DMEM lacking both phenol red or L-glutamine was supplemented with N-acetylcysteine (4mM, or 10mM) and added to 293T cells transiently transfected with dCas9-KR and sgAAVS1 and incubated at 37˚C for 1 hour prior to LED light treatment. Cells were exposed to green LED at 20K lux for 10 minutes or 1 hour and then placed at 37˚C for 15 minutes before formaldehyde fixation and collection for ChIP. The mean of four biological replicates is graphed with PCR technical replicate error bars.

## Western blot analysis

293T cells with or without the dCas9-KR and sgAAVS1 constructs were lysed in RIPA buffer (50mM Tris-HCl, pH 8.0; 150mM sodium chloride; 1.0% NP-40; 0.5% sodium deoxycholate; 0.1% sodium dodecyl sulfate) with 1X protease inhibitor cocktail (Roche, IN) and sonicated in a bioruptor for 2.5 mins in 30 second pulses at 4˚C. Debris was pelleted at 10,000 rpm and lysates (20 μg) separated in SDS-PAGE gels and semi-dry transferred onto nitrocellulose using standard methods. Primary antibodies (diluted in 5% milk) used were: anti-γH2AX (Abcam ab11174; 1:5000), anti-KillerRed (Evrogen AB961; 1:5000), and anti-β-actin (Cell Signaling Technologies 4967; 1:2000). Bands were detected with anti-rabbit secondary antibody (LI-COR; 1:10,000) and scanned on an Odyssey imager to quantify band intensity.

## Next generation sequencing

293T cells transiently transfected with dCas9-KR +/- sgAAVS1 were treated with green LED light at indicated intensities and times, 24 post-transfection. After treatment, the medium was changed to DMEM with 1 ug/ml puromycin to enrich for cells that maintained the dCas9-KR vector. At 48 hours post-treatment, cells were harvested and DNA purified using the Blood and Tissue DNA kit (Qiagen, MD). A 241 base pair amplicon spanning the AAVS1 target site was amplified in 25 cycles using NEB One Taq (AAVS1 -80bp Forward 5′ GACCACCTTA TATTCCCAGG; AAVS1 +161 bp Reverse 5′ GAGGTTCTGGCAAGGAGAGA) and purified using the PCR clean up kit (Qiagen, MD). Amplicons were sequenced by MiSeq (Illumina) at the CCIB DNA Core Facility at Massachusetts General Hospital (Cambridge, MA) using a MiSeq v2 chemistry 300 cycle kit. High-throughput analysis of amplicon deep sequencing was performed as in [79]. Briefly, paired end sequencing raw reads were trimmed to primer sequences and merged into single reads using Geneious v10.1.3. Only sequences with >20 bp of reference sequence adjacent to the primer sequence were analyzed. SAM files of sequences trimmed to common start and end sequences were exported using Geneious v10.1.3. HiFiBR [80] was used to classify sequences as exact, deletion, insertion, or complex (contains both insertion and deletion), with a threshold set at ≥10 reads. Each class of "repair" was expressed as a percent of events divided by total sequence reads. The average of two biological replicates is plotted.

## GFP to BFP conversion assay

293T cells were engineered to contain the GFP array as described in [39] and stable, GFP positive cells (<0.01% BFP positive) were used in subsequent experiments. In 6-well plates, 1 x 10^5 GFP+ 293T cells were seeded in DMEM complete medium 24 hr prior to transfection. dCas9-KR (1μg), AAVS1 sgRNA (1μg), and/or BFP template (50ng) were transfected with lipofectamine 2000 according to the manufacturer's instructions. The Cas9/AAVS1 sgRNA single vector (1μg) (Genecopoiea, MD) was used as a positive control. The BFP template is a 290 bp PCR fragment created by amplification of a custom G-block (Integrated DNA

Technologies, IA). G-block and primer sequences in S1 File). At 24 hr post-transfection, media was changed to DMEM lacking phenol red, 25μM olaparib added as required, and incubated for a further 2 hr at 37˚C. Cells were then exposed to 30K lux green LED for 1 hr. After treatment, cells were allowed to recover for 1 hr at 37˚C and then the media was replaced with fresh DMEM. Cells were grown for five days post-treatment, split 1:2, and grown for an additional two days. A Cytoflex flow cytometer was used to score percent of GFP$^+$ and BFP$^+$ cells.

## Southern blotting

DNA was isolated using the Qiagen Blood and Tissue DNA kit (Qiagen, MD) and digested with EcoRI-HF and Mfe1-HF (New England Biolabs, MA) to release a 3594 bp fragment surrounding the Cas9/p84 target site in AAVS1. Phenol:chloroform:isoamyl alcohol (25:24:1) extracted and ethanol precipitated DNA (20 μg) was run in 1% agarose and Southern transferred (standard procedures) onto Biodyne B nylon membranes (Thermo Scientific, MO). Blots were probed with a 753 bp PCR fragment spanning the AAVS1 site, from -80 bp to +673 bp (AAVS1 -80bp Forward 5' CTTGCTTTCTTTGCCTGGAC; AAVS1 +0.5kb Reverse 5' CGGAGGAATATGTCCCAGATAGCA), amplified with biotin-16-dUTP (Roche, IN). The biotin-labeled probe was detected with IRDye 800CW Streptavidin (LI-COR) and scanned on an Odyssey imager.

## Supporting information

**S1 Fig. Experimental set-up for illumination and KillerRed activation in cells expressing dCas9-KR.** (**A**) Photograph of actual apparatus for light delivery. The SugarCUBE LED with a green bulb is attached via a liquid light guide to an aluminum foil protected Styrofoam box. Adherent cells in plates were placed inside the box with the lid closed and the LED was turned on for the times indicated in individual experiments. Light intensity was measured using a luxometer placed underneath the plate of cells. Intensity was measured in lux, the SI unit of illuminance which is a measure of the amount of light emitted per second per square meter. (**B**) Typical experimental schematic. 293T cells were transiently transfected with the dCas9-KR and/or the AAVS1 sgRNA. At 24hr post-transfection, cells were illuminated by exposure to the SugarCUBE LED and then placed back in the incubator at 37˚C for recovery before cell collection. Illumination and recovery times varied by experiment. (**C**) 293T cells were transiently transfected with either vector (-), dCas9-KR or AAVS1 gRNA as indicated. 24 hrs later, cells were exposed to 20K Lux for the indicated times, followed by western blot analysis to detect dCas9-KR and b-actin (loading control).
(TIF)

**S2 Fig. ChIP for APE1, XRCC1 and H2AX at the dCas9-KR binding site.** (**A**) ChIP for APE1 following transfection of dCas9-KR plus the indicated sgRNA. ChIP was carried out using APE1 antibody and primer pairs at the indicated distance from the dCas9-KR site (located at 0 base pairs). $^* = p < 0.05$; $ns = non$-$specific$. (**B**) As in (A), but using XRCC1 antibody. (**C**) as in (A), but using H2AX antibody to monitor relative H2AX occupancy at the indicated positions. All methods described in materials and methods section.
(TIF)

**S3 Fig. Next generation sequence analysis.** Wild type or Ligase IV$^{-/-}$ 293T cells were transfected with vector, nuclease proficient Cas9 plus gRNA or dCas9-KR. dCas9-KR cells were illuminated at 1hr at 20K lux. DNA was isolated 48 hr post-illumination and a 241 bp fragment surrounding the dCas9-KR was amplified and sequenced by NGS. (**A**) Total misalignments compared to the reference sequence (non-transfected cells); (**B**) deletions; (**C**) insertions; (**D**)

DNA was isolated at the indicated time points post-illumination (0–120 minutes at 20K lux), followed by Southern blot to measure chromosome breakage at the AAVS1 site (left panel). The p84-ZFN nuclease was used as a positive control for DSB-induction (right panel, arrows). NT = not treated. The AAVS1 site was detected using a biotin-16-dUTP labeled amplicon spanning the AAVS1 site.
(TIF)

**S4 Fig. GFP to BFP repair assay to detect DNA damage induced by dCas9-KR.** In this assay, cells contain a GFP array that can be targeted by the dCas-KR construct. Frameshift mutations after repair convert GFP$^+$ to GFP$^-$ and this is used as a measure of NHEJ efficiency after DNA damage at a GFP array. When a BFP template is provided, repair can proceed via homologous recombination, resulting in gene conversion from GFP$^+$ to BFP$^+$. Inefficient DNA damage induction or infrequent mutational repair can result in GFP+BFP+ cells due to several copies of GFP being present in the target array. We have therefore scored any BFP+ cell as having undergone gene conversion/HR. The mean and error for at least three biological replicates are plotted. **(A)** Conversion from GFP$^+$ to GFP$^-$ to measure NHEJ after dCas9-KR ROS-induced DSB repair in untreated (0 lux) or illuminated (30K lux, 1 hr) cells. The percent of GFP$^-$ cells are plotted to represent cells that underwent NHEJ. **(B)** Nuclease proficient Cas9 constructs were used as positive controls. Cas9 and the guide RNA were either in the same vector (Cas9-gRNA) or co-transfected as separate vectors (Cas9, GFP sgRNA). **(C)** Conversion to BFP$^+$ to measure HR frequency after dCas9-KR ROS-induced DSB repair, performed as in (A). **(D)** Nuclease proficient Cas9 used as positive controls as in (B). **(E-H)** GFP to BFP conversion assay in the presence of Olaparib. Given that the dCas9-KR-induced DNA damage appeared to be quickly repaired (Figs 2E, 2F and 3E), we attempted to increase the frequency of DSBs by using olaparib, which traps PARP on the DNA, potentially limiting BER repair and causing DSBs (40, 41). Cells containing the dCas9-KR and GFP-targeted sgRNA were pre-treated in 25 uM Olaparib for two hours prior to illumination, and Olaparib was removed one hour post-illumination. For comparison, nuclease proficient Cas9 plus gRNA were also used. **(E)** dCas9-KR activation in Olaparib treated cells did not increase NHEJ (GFP$^-$) rates, and therefore NHEJ is relatively unaffected or occurring at levels that are too low to be detected by this assay. **(F)** Nuclease proficient Cas9 increased NHEJ, but was not altered by olaparib. **(G)** dCas9-KR activation appeared to increase HR (BFP+) frequency in Olaparib treated cells; however, the rates were variable and did not reach statistical significance even after several trials. **(H)** Nuclease proficient Cas9 control demonstrating increased HR mediated repair. **Conclusion:** We conclude that Olaparib treatment reveals that DNA damage is occurring at the dCas9-KR target site, and in a subset of cells these lesions are converted to DSBs (or lesions that initiate mutagenesis leading to GFP–or BFP+ in this assay) after PARP inhibition. However, dCas9-KR activation is inducing only a modest amount of DNA damage overall. Taken with our sequencing analysis (Fig 4B, 4C), this suggests that targeted dCas9-KR induced damage is easily repaired, even with transient BER inhibition.
(TIF)

**S5 Fig. Original western blot images from Fig 1.**
(TIF)

**S1 File.**
(DOCX)

**S1 Data.**
(XLSX)

**S2 Data.**
(XLSX)

**S3 Data.**
(XLSX)

**S4 Data.**
(XLSX)

## Acknowledgments

We thank Dany Spencer Adams for discussions regarding light delivery apparatus and intensity measurements, and members of the Price lab and Day lab for valuable discussions.

## Author Contributions

**Conceptualization:** Nealia C. M. House, Brendan D. Price.

**Formal analysis:** Ramya Parasuram, Jacob V. Layer.

**Funding acquisition:** Brendan D. Price.

**Investigation:** Nealia C. M. House, Ramya Parasuram.

**Methodology:** Nealia C. M. House.

**Supervision:** Brendan D. Price.

**Writing – original draft:** Nealia C. M. House.

**Writing – review & editing:** Nealia C. M. House, Brendan D. Price.

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
