## [Decision Letter · Decision Letter 0]

26 Aug 2020

PONE-D-20-23466

Site-specific targeting of a light activated dCas9-KIllerRed fusion protein generates transient, localized regions of oxidative DNA damage.

PLOS ONE

Dear Dr. Price,

Thank you for submitting your manuscript to PLOS ONE. After careful consideration, we feel that it has merit but does not fully meet PLOS ONE’s publication criteria as it currently stands. Therefore, we invite you to submit a revised version of the manuscript that addresses the points raised during the review process.

As noted in the reviewer comments, there are numerous small and important concerns raised that should be addressed. 

We look forward to receiving your revised manuscript.

Kind regards,

Robert W Sobol, PhD

Academic Editor

PLOS ONE

Journal Requirements:

"The authors have declared that no competing interests exist"

We note that one or more of the authors are employed by a commercial company: eGenesis.

2.1. Please provide an amended Funding Statement declaring this commercial affiliation, as well as a statement regarding the Role of Funders in your study. If the funding organization did not play a role in the study design, data collection and analysis, decision to publish, or preparation of the manuscript and only provided financial support in the form of authors' salaries and/or research materials, please review your statements relating to the author contributions, and ensure you have specifically and accurately indicated the role(s) that these authors had in your study. You can update author roles in the Author Contributions section of the online submission form.

2.2. Please also provide an updated Competing Interests Statement declaring this commercial affiliation along with any other relevant declarations relating to employment, consultancy, patents, products in development, or marketed products, etc.  

Reviewers' comments:

Reviewer's Responses to Questions

**Comments to the Author**

1. Is the manuscript technically sound, and do the data support the conclusions?

Reviewer #1: Partly

Reviewer #2: Yes

2. Has the statistical analysis been performed appropriately and rigorously? 

Reviewer #1: No

Reviewer #2: No

3. Have the authors made all data underlying the findings in their manuscript fully available?

Reviewer #1: Yes

Reviewer #2: Yes

4. Is the manuscript presented in an intelligible fashion and written in standard English?

Reviewer #1: Yes

Reviewer #2: Yes

5. Review Comments to the Author

Reviewer #1: House et al. utilize a dCas9-KillerRed expression construct to investigate double-strand break repair generated at the AAVS1 locus within cells. Using a 532nm LED source to generate ROS using KR, the authors investigate the repair mechanism for the ROS-generated DNA damage using ChIP, sequencing, and viability assays. While the ability to generate site-specific, transient DNA damage to identify error-rates of NHEJ is of interest, there are a number of issues with the manuscript.

The main issue is the use of the term ‘site-specific’. The novelty presented in this manuscript is the fusion of dCas9 to KR to introduce site-specific damage to the AAVS1 locus using the site-specific gRNA. However, not all dCas9-KR in the nucleus can enrich onto genomic DNA due to limited Cas9 loading onto specific genomic loci, thereby leading to non-specific oxidative DNA damage of unbound dCas9-KR. When the 532nm light is used, it will stimulate KR independent of dCas9 binding, resulting in ROS generation throughout the nucleus, similar to genotoxins such as H2O2. To show site-specific DNA damage, you will need to perform immunocytochemistry using �H2AX labeling to show focal �H2AX around the AAVS1 site. Additionally, Figure 1C hints at possible increases in global �H2AX, even when the sgRNA is not present, but no quantitation is provided. This should be repeated to include at least 3 replicates and semi-quantitative analysis of band intensity to bolster the author’s statement that there are no detectable global increases and the system is inducing site specific DNA damage. Otherwise, the system cannot be characterized as site-specific ROS generation but rather genomic locus-enriched ROS generation.

Other issues:

1) Figure 2C should include APE1 data for dCas9-KR without AAVS1 gRNA at 20K instead of the 5K provided.

2) Is the difference in template integrity statistically different in Figure 4A? If not, it should be addressed.

3) Figure 2B legend and text says that �H2AX was determined after 10 and 60 minutes, but the x-axis is labeled for 0, 10, and 30 minutes.

4) On page 11, line 219- authors state that “Light activation of dCas9-KR did not increase repair by either NHEJ…” when it should read “Light activation of dCas9-KR did not increase error-prone repair by NHEJ or…”

5) Figure 4 E and F are labeled as dCas9-KR when they should be Cas9.

Reviewer #2: The authors have presented a very clear and well written study characterizing their newly developed dCas9 fused to killer red. This allows the authors to generate targeted bursts of reactive oxygen species when excited with green light. This clustered oxidative damage is quickly and efficiently repaired and offers a more temporally controlled option for creating oxidative DNA damage when compared to other techniques that require induction or expression. Interestingly, this damage recruits both NHEJ and BER proteins, but does not seem to involve HR proteins. Furthermore, repair is very high fidelity as demonstrated by NGS and GFP to BFP gene conversion assays. This is uncharacteristic of normal NHEJ, indicating perhaps there is a coordinated repair effort between both pathways in the case of clustered oxidative lesions.

Overall, this seems to be an exciting and useful tool for further investigating DNA repair of oxidative lesions and is a valuable contribution to the field. The characterization experiments in this study are well designed and well carried out with appropriate controls throughout. The clarity and quality of writing was excellent. My overall recommendation is to accept with the following minor revisions.

Minor Revisions:

While most of the data presented has very clear differences that leave no question about the effect, the manuscript could be improved with the addition of statistical analysis in a few figure panels and a methods section describing that analysis.

Specifically:

Figure 2 Panel A and C

Figure 3 Panel E

Figure 4 Panels D and E

Figure 4 Panels F and G appear to have statistics represented but there is no discussion of p value represented by the * in the legend or indication of how statistical analysis was carried out.

6. PLOS authors have the option to publish the peer review history of their article (what does this mean?). If published, this will include your full peer review and any attached files.

Reviewer #1: No

Reviewer #2: No

---

## [Author Response · Author response to Decision Letter 0]

16 Nov 2020

Response to reviewers.

Reviewer 1.

Reviewer 1 requested additional clarification of the conclusion that dCas9-KR generates site-specific DNA damage, rather than general (locus specific) DNA damage due to ROS production throughout the nucleus. To address this, as requested, we repeated the gH2AX experiments in figure 1, using 3 biological replicates, with image quantification. This data is now included in figure 1, with the replicate blots used to generate the data presented in supplementary figure 5. This clearly shows that dCas9-KR activation does not lead to detectable increases in global gH2AX. Further, the ChIP data (figure 1) shows that gH2AX is only increased at the AAVS1 locus when both the targeting gRNA and light exposure are used, indicating that this damage requires docking of the dCas9-KR onto the chromatin to produce detectable damage. Although looking for specific gH2AX foci at the AAVS1 site would be a useful approach, in our initial characterization of the dCas9-KR system, we were unable to reproducibly detect gH2AX foci directly at the AAVS1 site. We believe this is due to the low gH2AX signal generated by dCas9-KR and the limited spreading of gH2AX from the site. However, ChIP, which is a more sensitive approach (figure 1), clearly shows the presence of increased gH2AX at the dCas9-KR locus. Taken together, we believe that this supports our original conclusion that dCas9-KR creates site-specific rather than global DNA damage.

(i) The original figure 2C was, in fact, incorrectly labelled. Since the changes in APE1 are small and were shown in more detail in the original supplementary figure 2, we have removed figure 2C and limit our discussion of this to the APE1 data set originally presented (supplementary figure 2A).

(ii) The difference in template integrity in figure 4A is not statistically significant. This was not made clear in the original submission, and we have provided statistical analysis in figure 4A and the figure 4 legend, as well as altered the text (lines 192-194) to reflect this.

(iii) The legend and figure 2B have been corrected to reflect that the exposure was 0, 10 and 60mins.

(iv) We have corrected line 219 (line 240 in the corrected version) and added “error-prone”.

(v) We have checked the labeling of figure 4E and figure 4F and these are now correctly annotated.

Reviewer 2.

We appreciate reviewer 2’s enthusiasm for the paper and thank them for their strong endorsement. 

We have made the following changes to the statistical analysis, as requested:

• We have added statistical analysis to Figure 2, Figure 3E (now figure 3D in revised version) and Figures 4D and 4E, as requested. In figure 2C, for clarity, statistical significance is noted in the figure legend rather than on the figure.

• Figure 2C in the original submission has now been removed, as noted above (point (i)), with the data now presented in supplementary figure 2.

• A description of the statistics in Figure 4D, 4E, 4F and 4G has been added to the figure legend.

---

## [Decision Letter · Decision Letter 1]

1 Dec 2020

Site-specific targeting of a light activated dCas9-KillerRed fusion protein generates transient, localized regions of oxidative DNA damage.

PONE-D-20-23466R1

Dear Dr. Price,

We’re pleased to inform you that your manuscript has been judged scientifically suitable for publication and will be formally accepted for publication once it meets all outstanding technical requirements.

Kind regards,

Robert W Sobol, PhD

Academic Editor

PLOS ONE

Additional Editor Comments (optional):

Reviewers' comments:

Reviewer's Responses to Questions

**Comments to the Author**

1. If the authors have adequately addressed your comments raised in a previous round of review and you feel that this manuscript is now acceptable for publication, you may indicate that here to bypass the “Comments to the Author” section, enter your conflict of interest statement in the “Confidential to Editor” section, and submit your "Accept" recommendation.

Reviewer #1: All comments have been addressed

Reviewer #2: All comments have been addressed

2. Is the manuscript technically sound, and do the data support the conclusions?

Reviewer #1: Yes

Reviewer #2: Yes

3. Has the statistical analysis been performed appropriately and rigorously? 

Reviewer #1: Yes

Reviewer #2: Yes

4. Have the authors made all data underlying the findings in their manuscript fully available?

Reviewer #1: Yes

Reviewer #2: Yes

5. Is the manuscript presented in an intelligible fashion and written in standard English?

Reviewer #1: Yes

Reviewer #2: Yes

6. Review Comments to the Author

Reviewer #1: (No Response)

Reviewer #2: (No Response)

7. PLOS authors have the option to publish the peer review history of their article (what does this mean?). If published, this will include your full peer review and any attached files.

Reviewer #1: No

Reviewer #2: No

---

## [Editor Report · Acceptance letter]

9 Dec 2020

PONE-D-20-23466R1 

Site-specific targeting of a light activated dCas9-KillerRed fusion protein generates transient, localized regions of oxidative DNA damage. 

Dear Dr. Price:

I'm pleased to inform you that your manuscript has been deemed suitable for publication in PLOS ONE. Congratulations! Your manuscript is now with our production department. 

Kind regards, 

on behalf of

Dr. Robert W Sobol 

Academic Editor

PLOS ONE